# Towards Automated Banff Lesion Scoring: Tissue Segmentation in Kidney Transplant Biopsies using Deep Learning

**Sebastiaan Ram**                                    SEBASTIAAN.RAM@RADBOUDUMC.NL

**Dominique van Midden**                    DOMINIQUE.VANMIDDEN@RADBOUDUMC.NL

**Jeroen van der Laak**                        JEROEN.VANDERLAAK@RADBOUDUMC.NL

**Linda Studer**                                    LINDA.STUDER@RADBOUDUMC.NL

*Diagnosic Image Analysis Group*
*Radboud University Medical Center*

## Abstract

Inflammation and chronic changes in the different tissue structures (e.g., glomeruli, tubuli, interstitium) are major contributors to kidney transplant failure. Kidney transplant biopsy diagnostics is based on the Banff classification system, in which pathologists assess these changes. However, many of these factors have suboptimal reproducibility and the scoring is labor-intensive. To address this, we developed a multi-class segmentation approach that covers all tissue structures relevant for diagnostics. Our dataset comprises 99 Periodic-acid Schiff (PAS)-stained kidney transplant biopsy slides from two pathology departments. An expert pathologist manually annotated >17,000 structures across eight classes (glomeruli, sclerotic glomeruli, empty Bowman space, proximal tubuli, distal tubuli, atrophic tubuli, capsule, arteries/arterioles, and interstitium). We compared two segmentation approaches: (1) a combination of two nnU-Nets (one for tissue segmentation and one specialized for structure boundary detection) and (2) the SAM-Path foundation model. For the peritubular capillary segmentation, we used a previously developed U-Net. The nnU-Nets achieved a per-class average Dice score of 0.80, outperforming SAM-Path (0.69) and providing a reliable solution for all tissue structures relevant for kidney transplant biopsy diagnostics. Next, the nnU-Nets will be used in a reader study aimed at investigating the impact of AI on pathologists' performance in Banff lesion scoring. The algorithm is publicly available on Grand Challenge[1].

**Keywords:** segmentation, nnUNet, segment anything model (SAM), kidney transplant biopsies, deep learning, Banff classification

## 1 Introduction

Digital pathology has rapidly advanced through the integration of deep learning, enabled by whole-slide imaging technologies that digitize biopsies into gigapixel images, so-called whole-slide images (WSIs). This shift allows computational analysis of tissue, opening new opportunities in workflow integration and disease diagnosis, including cancer and kidney pathology (Achi et al., 2019; Song et al., 2023).

---

1. https://grand-challenge.org/algorithms/kidney-tissue-segmentation

The kidneys are vital for filtering waste, regulating blood pressure, and maintaining electrolyte balance (Lopez-Giacoman and Madero, 2015; Luyckx et al., 2024). Chronic kidney disease affects approximately 10% of the global population and can progress to end-stage renal disease (ESRD), where kidney function declines irreversibly and renal replacement therapy becomes necessary (Lopez-Giacoman and Madero, 2015). Kidney transplantation is a common treatment for ESRD, which is often caused by diabetes, hypertension, or polycystic kidney disease (Hariharan, 2001). Despite improvements in surgical techniques and immunosuppressive therapies, long-term graft survival remains challenged by chronic rejection and inflammation (Hamed et al., 2015; Heldal et al., 2023). Histological analysis of tissue compartments, including the interstitium, tubuli, glomeruli, and peritubular capillaries, is central to assessing graft rejection.

The Banff classification provides an internationally standardized system for grading transplant rejection. It defines criteria for inflammation and chronic changes such as interstitial fibrosis (ci), tubular atrophy (ct), tubular atrophy with fibrosis (i-IFTA), and peritubular capillaritis (ptc), using categorical scales from zero (none) to three (severe) (Naesens et al., 2024). Banff Lesion Scores are derived from these criteria and play a central role in transplant diagnostics and treatment planning.

Convolutional neural networks (CNNs) have shown strong performance in segmenting kidney structures. For example, Hermsen et al. (2022) used a U-Net-based architecture to segment major tissue compartments, achieving high accuracy on PAS-stained biopsies. More recent models such as Omni-Seg (Deng et al., 2023) and PrPSeg (Deng et al., 2024) incorporate multi-resolution patch inputs to capture tissue features at different spatial scales. While these techniques have demonstrated improved segmentation performance, they require task-specific training pipelines and complex architectural modifications.

More recently, vision foundation models such as SAM-Path (Zhang et al., 2023) and SAM-Nephro (Weijer et al., 2024) have shown promise for domain-agnostic segmentation in digital pathology. Unlike traditional CNNs, these models can often be applied out-of-the-box, requiring minimal fine-tuning to perform meaningful segmentation. However, their effectiveness on PAS-stained kidney transplant biopsies has not been systematically evaluated. It should also be noted that SAM-Nephro is primarily designed as an instance segmentation–based annotation tool, with limited standalone segmentation capability.

Our contributions are threefold. First, we develop a multi-class structure segmentation model for kidney transplant biopsies using nnU-Net, an automated and highly adaptable segmentation framework that configures itself to a given dataset (Isensee et al., 2021). Second, we integrate contextual information from other structural compartments with the results from existing peritubular capillary (PTC) segmentations. Third, we compare the performance of dedicated CNN-based models with foundation model-based approaches for PAS-stained slides.

## 2 Methods

The nnU-Net framework (Isensee et al., 2021) is a self-configuring segmentation pipeline based on the original U-Net architecture. It automates preprocessing, architecture configuration, training, and postprocessing based on a dataset-specific fingerprint that encodes key properties such as spacing, modality, and intensity distribution. We used the

pathology-specific extension of nnU-Net proposed by Spronck et al. (2023), which addresses domain-specific challenges such as gigapixel-scale resolution and multi-class segmentation of morphologically diverse tissue structures.

SAM-Path (Zhang et al., 2023) is a pathology-adapted extension of the Segment Anything Model (SAM), a foundation model that enables prompt-free semantic segmentation. It augments the standard SAM architecture by introducing a parallel pathology encoder, whose features are combined with those from SAM's image encoder and passed to a shared decoder. A set of trainable class prompts replaces manual input, allowing the model to generate one segmentation mask per tissue class automatically.

All models were trained using a combination of Dice loss and focal loss. The total loss was computed as:

$$\mathcal{L}_{total} = w_{DC}\mathcal{L}_{Dice} + w_{Focal}\mathcal{L}_{Focal}, \tag{1}$$

where $w_{DC}, w_{Focal} \in [0, 1]$ denote the respective weights assigned to each loss term.

The Dice loss directly optimizes spatial overlap and is effective for sparse classes:

$$\mathcal{L}_{Dice} = -\frac{2 \sum Q_t Y_t + \epsilon}{\sum Q_t + \sum Y_t + \epsilon}, \tag{2}$$

where $Q_t$ and $Y_t$ denote the predicted and ground-truth binary masks, and $\epsilon$ prevents division by zero.

The focal loss (Lin et al., 2018) addresses class imbalance by down-weighting well-classified examples:

$$\mathcal{L}_{Focal} = -\log\left(\frac{N}{c_i}\right)(1 - p_t)^\gamma \log(p_t), \tag{3}$$

where $p_t$ is the predicted probability for the true class, $\gamma$ controls the focusing strength, and class weights are set via inverse class frequency (ICF), $\alpha_i = \log\left(\frac{N}{c_i}\right)$, with $c_i$ the pixel count of class $i$ and $N$ the total pixel count (excluding background). This formulation penalizes underrepresented classes (e.g., sclerotic glomeruli or atrophic tubuli) without inducing instability.

Finally, performance was assessed per class using the Dice score metric. Full computation details are provided in the supplementary materials (S1 and S2).

## 3 Experimental Setup

This study utilized 99 PAS-stained kidney transplant biopsy WSIs from Radboud University Medical Center (Radboudumc, $n = 69$) and Mayo Clinic, Rochester ($n = 30$). Slides were scanned using a PANORAMIC 1000 (3DHISTECH; Radboudumc) at a resolution of 0.24 µm/px and an Aperio ScanScope XT (Leica Biosystems; Mayo Clinic) at a resolution of 0.49 µm/px. An expert pathologist annotated eight classes: glomeruli, sclerotic glomeruli, empty Bowman's space, tubuli, atrophic tubuli, capsule, arteries/arterioles, and interstitium. Nearly 20,000 individual structures were annotated (Table 1).

Table 1: Overview of the number of annotated structures in each data subset for each tissue class.

| Tissue class | Train | Validation | Test | Total |
|---|---|---|---|---|
| Glomeruli | 140 | 55 | 76 | **271** |
| Sclerotic glomeruli | 121 | 47 | 6 | **174** |
| Empty Bowman's space | 73 | 14 | 10 | **127** |
| Tubuli | 7,921 | 2,637 | 3,240 | **13,798** |
| Atrophic tubuli | 3,240 | 899 | 368 | **4,507** |
| Capsule | 22 | 6 | 4 | **32** |
| Arteries/arterioles | 630 | 210 | 61 | **901** |
| Interstitium | 84 | 23 | 36 | **143** |

The dataset was split into training (47 Radboudumc, 16 Mayo), validation (12 Radboudumc, 4 Mayo), and test sets (10 Radboudumc, 10 Mayo), as in Hermsen et al. (2022). The pixel and class distributions per set are detailed in Table 1 and Figure S.1.

As an additional objective, 16 extra biopsies (13 Radboudumc, 3 Medical University of Vienna) were used to test whether the segmentation could improve the results of a previously developed PTC segmentation model (van Midden et al., 2025). These biopsies were annotated for PTC segmentation only, and slides were scanned using a PANORAMIC 1000 (3DHISTECH; Radboudumc) at 0.24 µm/px.

The nnU-Net was trained using $512 \times 512$ px patches at 1.0 µm/px resolution with 5-fold cross-validation, where fold assignments were optimized to match the dataset-wide class pixel distribution. The network was trained for 100 iterations with a batch size of eight, using batch normalization, LeakyReLU activation (slope 0.01), and a set of augmentations including spatial (scaling, rotation, mirroring), color (HSV, gamma), and deformation transforms. During inference, tissue-background segmentation (Bándi et al., 2019) masked non-tissue areas, followed by patch-wise sliding window prediction with half-overlap and Gaussian importance weighting (Isensee et al., 2021). To better resolve adjacent or overlapping structures, a separate border nnU-Net was trained to segment three classes: structure, border, and interstitium, using the same training configuration as the baseline model. Borders were generated by dilating ground truth masks by eight pixels (four inward, four outward) as illustrated by Figure 2.

SAM-Path was trained on $1024 \times 1024$ patches at 0.5 µm/px. We used the ViT-H SAM encoder pretrained on SA-1B, combined with a pathology-specific encoder: either HIPT (Chen et al., 2022) or UNI (Chen et al., 2024). Only the prompt and mask decoders were fine-tuned since the encoder weights were frozen. Training used a batch size of 12 and a learning rate of $1 \times 10^{-4}$ over 60 iterations. Class-balanced sampling and augmentations matching the nnU-Net were used for comparability. During inference, each class was prompted with a trainable token to generate confidence-weighted patch predictions, which were then stitched into full-resolution segmentation masks.

Both the nnU-Nets and SAM-Path used the AdamW optimizer during training, and an equal-weighted Dice and focal loss ($\mathcal{L}_{total} = \mathcal{L}_{Dice} + \mathcal{L}_{Focal}$) with $\gamma = 2$. Final mask

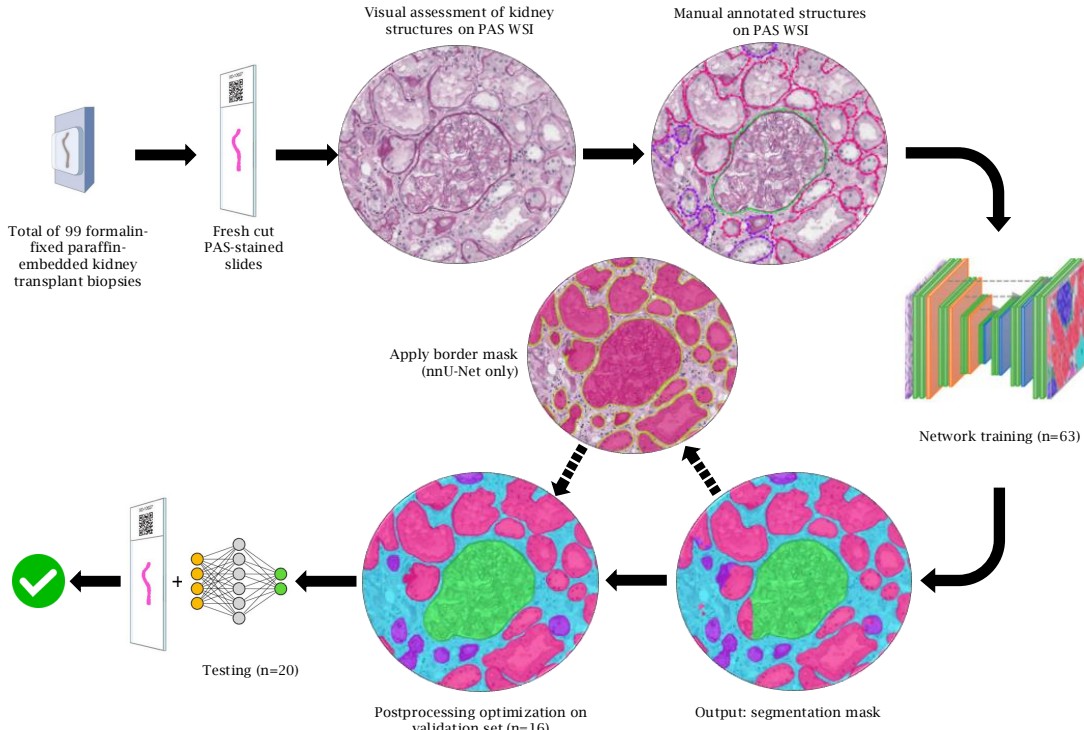

Figure 1: Overview of the nnU-Nets and SAM-Path training workflow: After staining and scanning, a pathologist annotated ground truth regions on 99 WSIs. The over 140 regions were split into training, validation, and test set. Postprocessing was applied after inference, utilizing the nnU-Net border network to refine the segmentation masks before postprocessing. These steps were iteratively optimized on the validation set, and final performance was evaluated on 20 hold-out test cases.

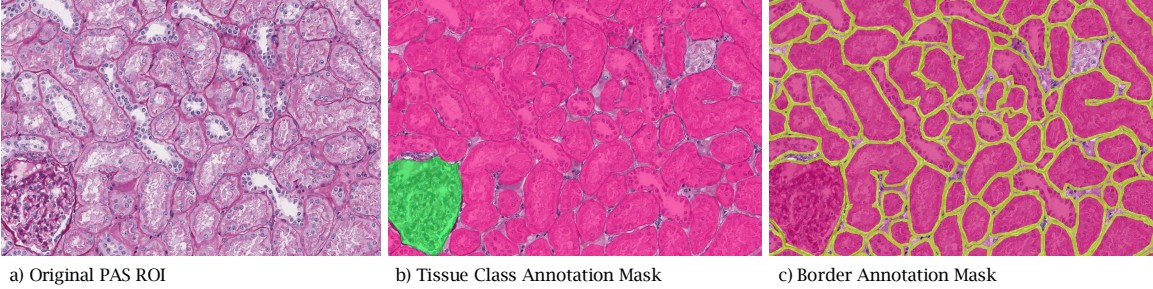

a) Original PAS ROI            b) Tissue Class Annotation Mask            c) Border Annotation Mask

Figure 2: Separating the different tissue compartments, especially tubuli, was challenging. To combat this, we trained a separate nnU-Net specifically for border segmentation (three classes: structure, border, and interstitium).

Table 2: Per-class Dice score of the best performing nnU-Nets (segmentation and border) and SAM-Path (UNI encoder) approach on the hold-out test set. The nnU-Nets outperform SAM-Path on all the tissue classes. Both models score below average for rarer structures, such as atrophic tubuli and empty Bowman's space.

|  | nnU-Nets | SAM-Path |
|---|---|---|
| Glomeruli | **.95** | .92 |
| Sclerotic glomeruli | **.76** | .29 |
| Empty Bowman's space | **.66** | .63 |
| Tubuli | **.92** | .87 |
| Atrophic tubuli | **.51** | .39 |
| Capsule | **.89** | .85 |
| Arteries/arterioles | **.82** | .71 |
| Interstitium | **.89** | .83 |
| **Weighted average** | **.74** | .47 |
| **Per-class average** | **.80** | .69 |

predictions were stitched to produce full-resolution segmentation masks. These masks then underwent postprocessing by temporarily setting interstitium pixels to zero, filling small holes ($<$150 pixels) with neighboring structure labels, and removing small objects ($<$300 pixels) as noise (reassigned to interstitium). Subregions within structures were merged into the dominant label. Finally, interstitium pixels were restored.

These models are still missing one tissue class, PTC, for a fully automated Banff scoring. We integrated this tissue class using an improved version of a previously developed model specifically for PTC segmentation (van Midden et al., 2025). Since PTCs are found in the interstitium, our prediction mask only includes PTCs that overlap with the predicted interstitial regions (see Figure 4).

## 4 Results and Discussion

The final models were chosen based on their performance on the validation set, which can be found in Table S.1. Table 2 presents the segmentation performance of the best-performing nnU-Net and SAM-Path approach on the hold-out test set. Per-class Dice scores are computed for each slide and then aggregated across all slides in the test set.

For the nnU-Net, we find that the model benefited greatly from extensive data augmentation and the combination with the border model. For SAM-Path, the UNI encoder configuration yielded the best results.

The nnU-Nets outperformed SAM-Path across all tissue classes, achieving a 0.80 average Dice per class versus 0.69 for SAM-Path, and a weighted average Dice of 0.74 compared to 0.47. The nnU-Nets' performance was also more consistent, particularly for common structures such as glomeruli, tubuli, capsules, and arteries/arterioles, as highlighted in Figure S.3. Both models struggled with rarer classes, such as atrophic tubuli and empty

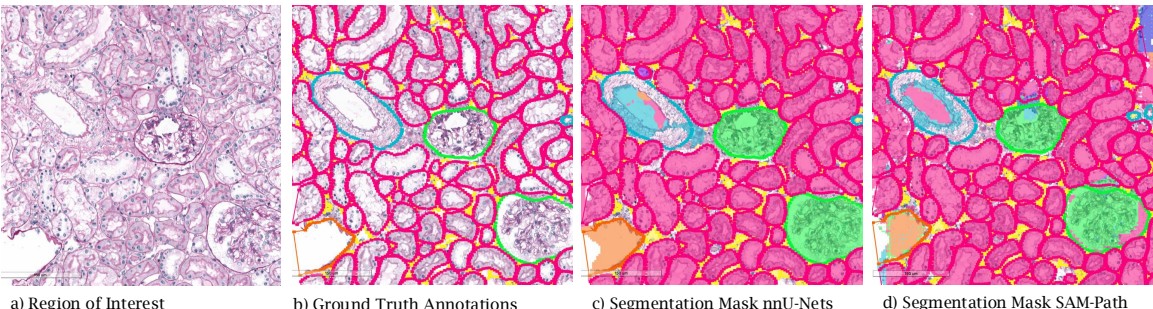

a) Region of Interest  b) Ground Truth Annotations  c) Segmentation Mask nnU-Nets  d) Segmentation Mask SAM-Path

Figure 3: Segmentation performance between **(c)** the nnU-Nets and **(d)** SAM-Path compared to **(b)** the ground truth (GT) annotations. The segmentations of both model outputs are aligned with the GT for reference.

Bowman's spaces. However, the nnU-Nets' Dice score for atrophic tubuli is 0.12 higher than SAM-Path.

Looking at the confusion matrix of the nnU-Nets (Figure S.2), we find that all classes except atrophic tubuli show a high Dice score. Atrophic tubuli are often misclassified as normal ones. Tubular atrophy is a continuous process, ranging from mild to severe. Often, only a part of a tubule is detected as atrophic, whereas the rest is labelled as normal. Examples of difficult cases for several tissue classes are shown in Figure S.5.

The border nnU-Net improved the separation of overlapping structures by learning border-specific features independently of class. This approach contrasts with integrated border-class strategies, reducing confusion between structures and their boundaries. However, segmentation quality still depends on accurate border detection, and missed borders may exclude structures during the merging process.

SAM-Path was evaluated using three encoder variants: the base SAM encoder and versions combined with HIPT or UNI. The pathology-specific encoders showed better results but were unable to perform on par with the nnU-Nets. This is likely due to these foundation models being pretrained on H&E- and IHC-stained WSIs, but not on PAS. Additionally, SAM-Path omits IoU-based mask selection, instead assigning pixels based on the highest class confidence, which may reduce robustness. Future work could reintroduce IoU scoring or adopt overlapping patch inference to address this.

Regarding runtime, training the nnU-Net required approximately 2 minutes per epoch, totaling around 3.5 hours per fold. In contrast, training SAM-Path's class prompts and mask decoder took approximately 5.5 hours each, with significantly higher GPU memory usage ($\sim 40$ GB), which limited the available batch size and hardware accessibility.

The integration of the PTC segmentation mask resulted in a 0.04 decrease in the PTC Dice score. Based on visual inspection, we found that some correctly predicted PTCs were removed because they were segmented as tubuli by the nnU-Nets and thus were not considered during the merge.

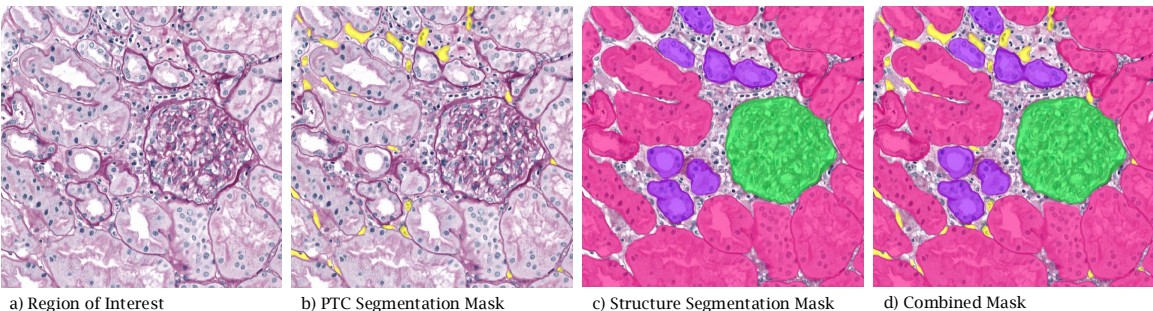

a) Region of Interest     b) PTC Segmentation Mask     c) Structure Segmentation Mask     d) Combined Mask

Figure 4: **(b)** PTC segmentations from a previous study (van Midden et al., 2025) were integrated to refine tissue classification **(c)**, with overlays limited to interstitial regions **(d)**.

## 5 Conclusion

This study developed and compared two approaches for tissue segmentation in PAS-stained kidney transplant biopsies: a task-specific nnU-Net model enhanced with a dedicated border segmentation network, and the foundation-based SAM-Path model augmented with pathology-specific encoders.

The nnU-Nets consistently outperformed SAM-Path, delivering reliable segmentation across all clinically relevant tissue classes. While SAM-Path showed promise on unseen data, its lack of training on PAS-stained images limited generalization. These findings underscore the importance of training foundation models on more diverse datasets encompassing multiple organs and stainings, including PAS-specific augmentations tailored to this domain.

To address the inseparability of structures during postprocessing, a separate border segmentation model was implemented to enhance the separation between adjacent structures. However, this occasionally led to the unintended removal of structures, suggesting that future work could explore joint training with border-aware loss functions.

The integration of PTC segmentation with the nnU-Nets has room for improvement, as it currently results in a decrease in the Dice score for the PTCs. Refining postprocessing by utilizing the model's prediction uncertainty or applying size-based constraints could improve this.

Finally, the segmentation of tissue structures is complicated by biological ambiguity and inter-observer variability, particularly for atrophic tubuli (Breda et al., 2020; Hermsen et al., 2022). From a clinical perspective, reliable identification of fibrotic versus normal tissue is more valuable than achieving perfect pixel-level accuracy.

For future work, we will conduct a reader study using this segmentation pipeline together with the inflammatory cell detection algorithm developed as part of the MONKEY challenge (Studer, 2024). The reader study will focus on how pathologists can use AI assistance for Banff lesion scoring and its impact on inter- and intra-observer variability.

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

## S1 Supplementary Materials: Software and Resources

All experiments were conducted on a deep learning cluster provided by Radboud University Medical Center. Each nnU-Net fold was trained on either an NVIDIA RTX 2080 Ti, RTX 3080 Ti, or GTX Titan X GPU. Training the SAM-Path mask decoder and class prompts required more virtual memory, necessitating the use of larger GPUs, specifically an NVIDIA A100 or L40S.

The experiments were implemented using Python (version 3.10) and PyTorch (version 2.6). For processing WSIs, we used the WholeSlideData package in Python[2]. The WSIs were annotated by professional pathologists using the automated slide analysis platform (ASAP, version 2.1, Computational Pathology Group; RadboudUMC[3])

Additionally, we used the publicly available repositories of nnU-Net for pathology (v1)[4] and SAM-Path[5] with minimal adjustments.

## S2 Supplementary Materials: Model Evaluation

This section briefly describes the evaluation metrics used and provides a detailed description of the performance of the nnU-Net border model.

### S2.1 Dice Score

Performance was assessed using the Dice score for each class. These metrics were computed from the confusion matrix $M \in \mathbb{N}^{C \times C}$, where $M_{ij}$ denotes the number of pixels of ground-truth class $j$ predicted as class $i$.

$$\text{Dice}_i = \frac{2M_{ii}}{\sum_j M_{ij} + \sum_j M_{ji}} \tag{4}$$

Where $i \in \{1, \dots, C\}$ corresponds to each of the eight tissue classes. For overall performance, the unweighted average of each per-class metric was reported:

$$S = \frac{1}{C} \sum_{i=1}^{C} s_i, \tag{5}$$

where $s_i$ denotes the Dice metric for class $i$. To emphasize rare class performance without overwhelming common class contributions, we also used normalized inverse class frequency (ICF) weighting:

$$ICF_{norm,i} = \frac{N}{nc_i}, \tag{6}$$

where $n$ is the number of classes and $c_i$ the pixel count for class $i$.

2. https://github.com/DIAGNijmegen/pathology-whole-slide-data

3. https://github.com/computationalpathologygroup/ASAP/releases/tag/ASAP-2.1

4. https://github.com/DIAGNijmegen/nnUNet-for-pathology/tree/nnunet_for_pathology_v1

5. https://github.com/cvlab-stonybrook/SAMPath

## S2.2  Evaluation of Border Network

The border network achieved an average Dice score of 0.84, with a weighted Dice score of 0.79, where the lower weighted score reflected occasional challenges in segmenting structure borders. Dice scores for borders and structures were 0.71 and 0.94, respectively. No post-processing was applied to the border predictions. However, small segmentations (<300 px) were filtered out during the merging stage with the structure masks.

## S2.3  Tables and Figures

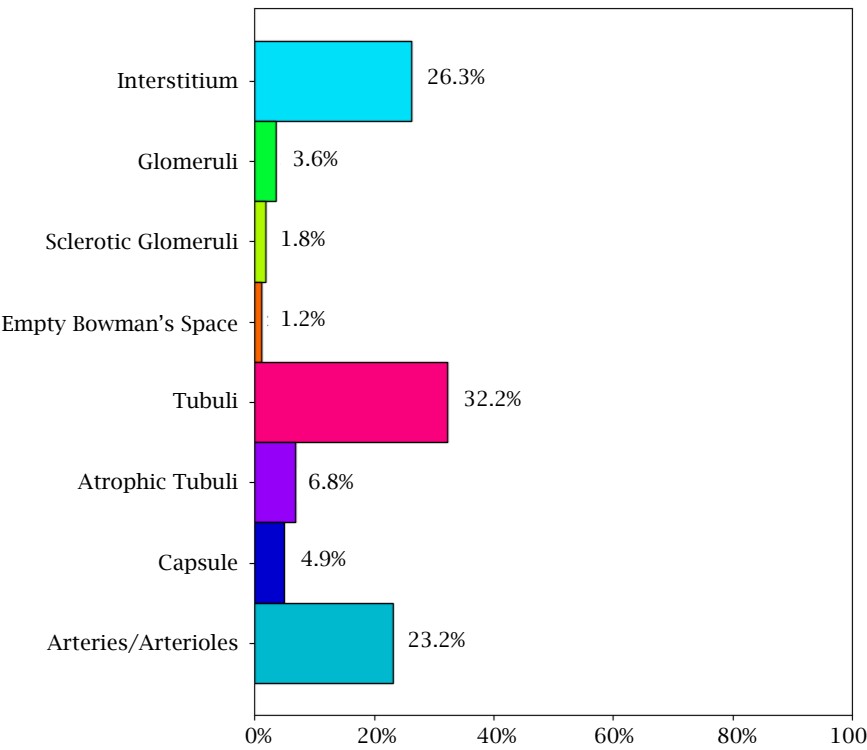

Figure S.1: Class pixel distribution of the 99 PAS-stained WSIs. A significant number of pixels is attributed to the tubuli, interstitium, and arteries classes, whereas fewer pixels correspond to empty Bowman's space, sclerotic glomeruli, glomeruli, capsule, and atrophic tubuli.

| | SAM-Path | | | nnU-Net | |
|---|---|---|---|---|---|
| | Base | HIPT | UNI | Standalone | With Border |
| Glomeruli | .519 | .766 | **.863** | .944 | **.956** |
| Sclerotic glomeruli | .309 | .621 | **.799** | **.926** | .924 |
| Empty Bowman's space | .445 | .486 | **.791** | .888 | **.923** |
| Tubuli | .752 | .792 | **.831** | **.902** | .899 |
| Atrophic tubuli | .398 | .458 | **.508** | **.657** | **.657** |
| Capsule | .434 | .690 | **.891** | .893 | **.925** |
| Arteries/arterioles | .587 | .741 | **.900** | **.951** | .933 |
| Interstitium | .705 | .731 | **.767** | .872 | **.874** |
| **Weighted average** | .438 | .593 | **.802** | .885 | **.905** |
| **Per-class average** | .518 | .661 | **.794** | .873 | **.886** |

Table S.1: Dice scores for segmentation predictions on the postprocessed validation set. For SAM-Path, three model configurations are evaluated: the base SAM-Path without an additional encoder, and SAM-Path with an added parallel encoder from either HIPT or UNI. For nnU-Net, the standalone model is compared to a version enhanced with a border segmentation network. Both nnU-Net configurations consistently outperform the SAM-Path models with the UNI encoder.

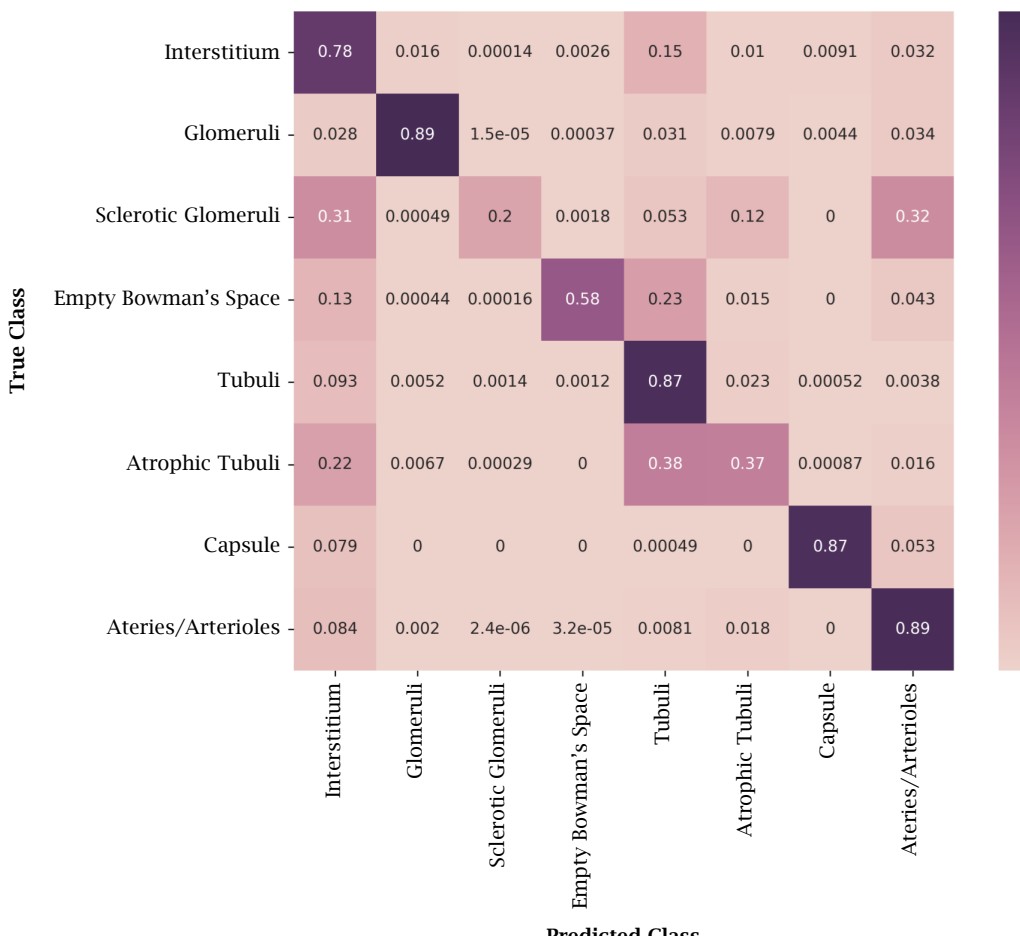

Figure S.2: Confusion matrix for the predictions of the nnU-Nets on the hold-out test set. The diagonal values represent correctly classified instances for each tissue class, while off-diagonal values indicate misclassifications. The model performs well on glomeruli, tubuli, capsules, and arteries/arterioles, with high per-pixel accuracy in these classes. However, atrophic tubuli are frequently misclassified as normal tubuli, and empty Bowman's space shows confusion with interstitium and glomeruli, highlighting challenges in distinguishing these structures.

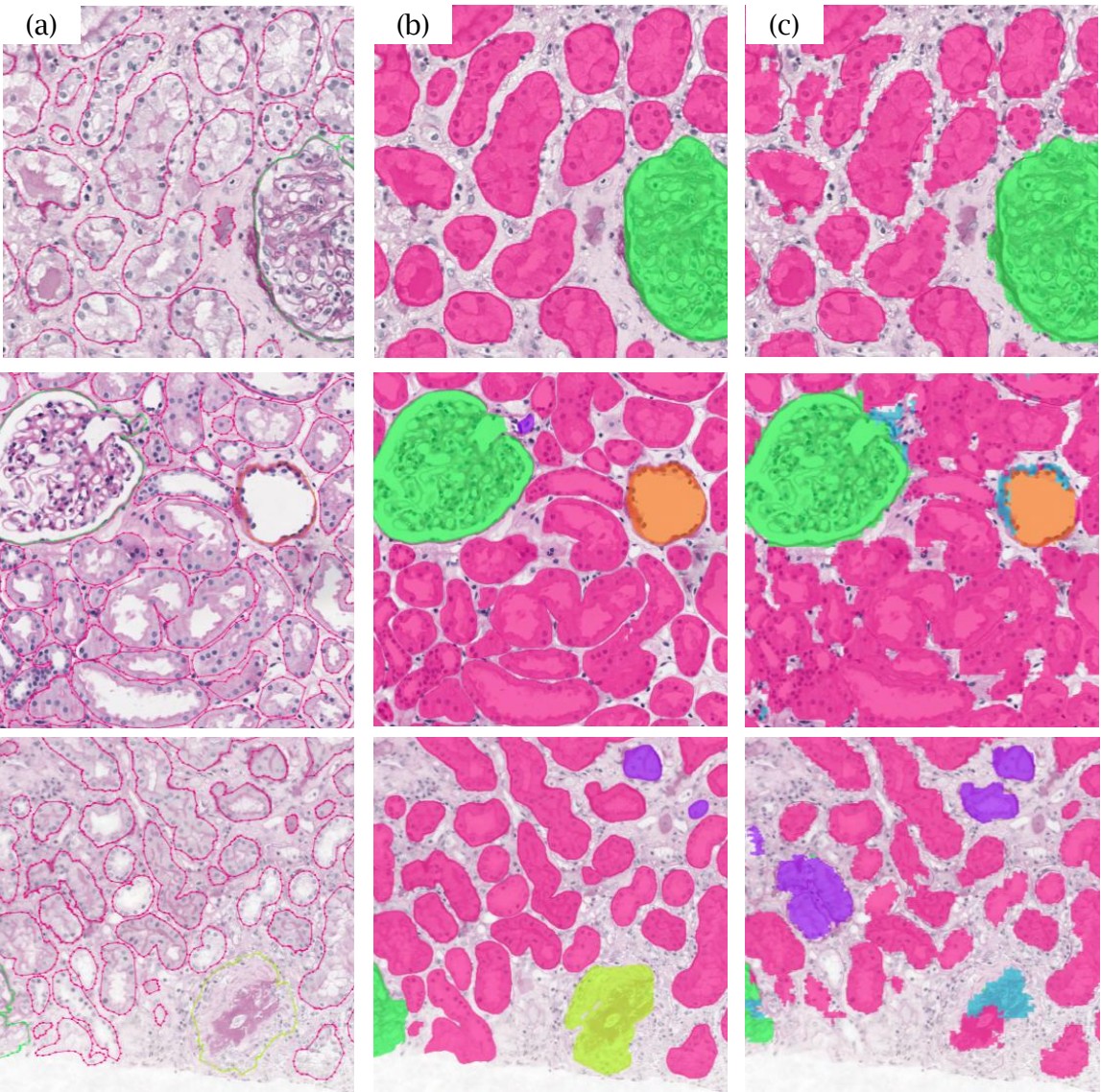

Figure S.3: Comparison of segmentation outputs between the nnU-Nets and SAM-Path with the UNI encoder. **(a)** Ground truth annotations, showing the manually labeled structures. **(b)** Predictions from nnU-Net, capturing most structures with high accuracy but showing some inconsistencies in finer details. **(c)** Predictions from SAM-Path with the UNI encoder, demonstrating differences in segmentation quality, particularly in small structures and boundary regions.

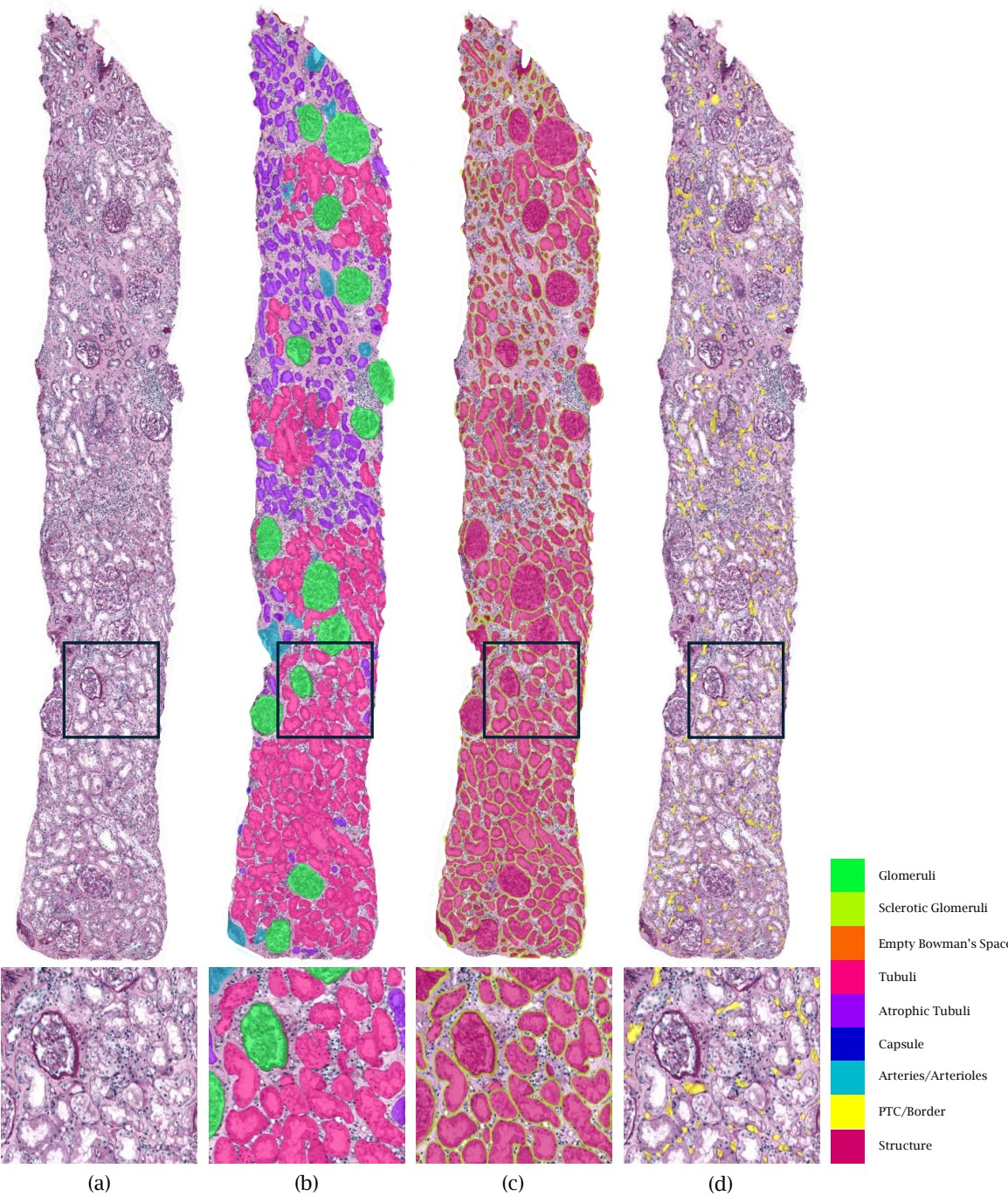

Figure S.4: Visualization of different segmentation masks applied to a kidney biopsy tissue. From left to right: **(a)** the original PAS-stained biopsy slide, **(b)** the structure segmentation mask highlighting various tissue structures, **(c)** the border mask showing boundaries between structures, and **(d)** the PTC segmentation mask. The bottom row presents a zoomed-in region of interest for each mask.

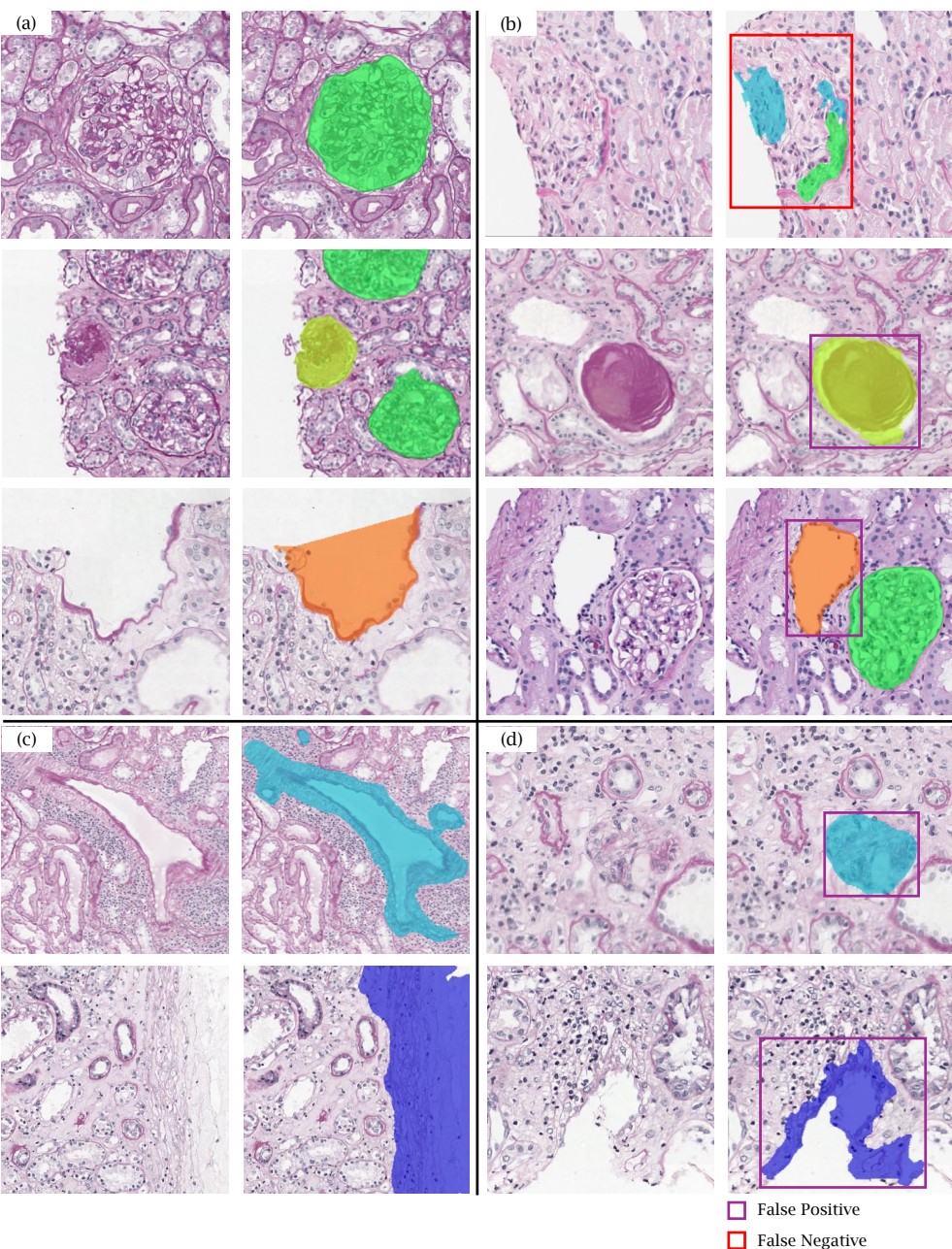

False Positive
False Negative

Figure S.5: Segmentation results and errors for different kidney structures from the best performing nnU-Net model. **(a)** Correctly segmented glomeruli, sclerotic glomeruli, and empty Bowman's space. **(b)** Incorrectly segmented structures, where purple boxes indicate false positives (regions predicted as non-structure with respect to the structure in the leftmost column), and red boxes indicate false negatives (missed structure regions). **(c)** Correctly segmented arteries and capsule structures. **(d)** Incorrectly predicted arteries and capsule structures, with false positives and false negatives highlighted in red and purple, respectively.

