# OpenReview forum: "Towards Automated Banff Lesion Scoring: Tissue Segmentation in Kidney Transplant Biopsies using Deep Learning"
_MICCAI.org/2025/Workshop/COMPAYL — COMPAYL 2025_

### Official Review · Reviewer_Lt7n · 2025-07-13
**Here, authors present a deep learning-based segmentation method designed to identify eight different kidney tissue structures for Automated Banff Scoring. The approach utilizes a nn-UNet-based architecture, followed by a dedicated boundary correction network to refine tissue boundaries.**

**Rating:** 4
**Confidence:** 4

**Review:**

Here, authors present a deep learning-based segmentation method designed to identify eight different kidney tissue structures for Automated Banff Scoring. The approach utilizes a nn-UNet-based architecture, followed by a dedicated boundary correction network to refine tissue boundaries. This method addresses an important clinical need in automated Banff scoring for kidney transplantation assessment, which could have significant practical implications for enhancing efficiency and consistency. The proposed segmentation method demonstrates several notable strengths that contribute to its potential value. The results appear promising for identifying multiple kidney tissue structures, suggesting that the approach is effective for its intended purpose. The combination of nn-UNet-based architecture with a dedicated boundary correction network presents an innovative approach to addressing the segmentation challenge, particularly in how it attempts to refine tissue boundaries for improved accuracy.

# Weaknesses

The manuscript suffers from several weaknesses that limit its impact and clarity. The introduction lacks clear articulation of how the proposed method addresses limitations of existing approaches, particularly regarding the statement about computational overhead and fully annotated datasets. It remains unclear how this method differs from other approaches with similar requirements (fully annotated datasets), which weakens the motivation for the proposed solution.

The paper would benefit from more comprehensive background information on kidney failure prevalence and significance, which would better frame the importance of kidney transplantation and accurate tissue segmentation. This missing context makes it difficult for readers to fully appreciate the clinical significance of the work.

The experimental validation also has several gaps. Most notably, there is no comparison with SAM-Nephro despite its relevance to kidney tissue segmentation. The authors also claim nn-UNet handles staining variability, but it requires a cross-center generalization, or slides scanned under different settings (e.g., training on one batch vs testing on another batch scanned under different settings) to support this assertion. Minor technical issues include the absence of exact values for loss function parameters such as w_dc, w_focal, gamma, and beta, which limits reproducibility. The mathematical notation in Equations (2) and (3) could be improved by using different symbols for probabilities and binary masks to enhance clarity.

## Below are my detailed comments aimed to further strengthening the manuscript:

# Major comments:
1.	The introduction can benefit from a clearer articulation of the motivation behind the proposed method. Specifically, the statement "While these techniques improve segmentation accuracy, they rely on fully annotated datasets and introduce additional computational overhead" could be elaborated further. It remains unclear how the proposed approach addresses this limitation or differentiates itself from existing methods that also rely on fully annotated datasets.
2.	I suggest including some background on kidney failure and its prevalence, which would help frame the significance of kidney transplantation as a treatment option. This would improve the flow into the sentence "Kidney transplantation is a common treatment for end-stage renal disease" and help ground the importance of accurate tissue segmentation.
3.	Could the authors clarify why their method was compared only with SAM-Path and was not compared with SAM-Nephro, especially after tuning it for kidney tissue segmentation? Given that SAM-Nephro is designed for a similar task, such a comparison would strengthen the experimental validation.
4.	The manuscript mentions that the proposed nn-UNet handles domain-specific challenges like staining variability. To support this claim, I suggest including an additional experiment demonstrating generalization across data from different centers (cross-center).
5.	The manuscript mentions both cross-validation and hold-out test set evaluation. If the model was trained using five different training splits and then evaluated on a separate hold-out set, please clarify this setup. Including details such as standard deviation would also provide more transparency and robustness to the reported results.
6.	Could you please clarify whether the results were computed by stitching the patches together to compare slide-level ground truths with slide-level predictions, or if the test set patches were evaluated independently of the slides they originate from? It might be beneficial to report patient level mean and standard deviation to show the generalizability of the approach.

# Minor comments:

1.	Please provide the values and their impact used for the loss function parameters:w_dc. w_focal, , gamma, and beta.
2.	For clarity, please consider using different symbols for probabilities and binary masks in Equations (2) and (3).
3.	It would be helpful to also mention whether the results presented in Table 2 were obtained before or after applying boundary correction while explaining the results (its mentioned in the figure).
4.	Please provide references for the nn-UNet model in the introduction

---

### Official Review · Reviewer_JDAz · 2025-07-15
**AI for Banff Scoring system**

**Rating:** 4
**Confidence:** 5

**Review:**

The paper focus on segmentation of tissue constitutes relevant for the classification of rejection grade in kidney transplant biopsies. The study compares performance between nnU-Net and SAM-Path.

Pros:
Interesting work and relevant problem. The study also includes data from multiple medical centers.

Cons(iderations):
- The way the model’s generalization to external cohorts is performed is confusing. What is referred to as external cohort includes data from Radboudumc medical center which is also part of the internal cohort? Why did you mix the data in this way? With 3 cohorts you could train/test the model on 2 of them and used the third for real external validation. Under the current set-up it can’t be called external validation. Also please specify how many samples from each medical center is used in the “external validation”.
- For the image resolution, can you please report also magnification, not only microns per pixel.
- Table 1, please use comma instead of period to report the numbers e.g. 1,000 instead of 1.000.
- The study applies boarder correction to nnU-Net but not for the SAM-Path predictions. Is this fair comparison? Why the correction was not performed for SAM-Path? You should also report scores from nnU-Net without correction to help determine if the improved scores from nnU-Net are due to border correction or due the model iteself, and to truly assess its benefits over SAM-Path
- The results in table 2, on which dataset/cohort are they presented? Is this the performance on external cohorts? It would be beneficial to include performance on truly external cohorts to allow comparison of the nnU-Net and SAM-Path, not only for model performance but also in terms of model generalization to external cohorts.
- How were the samples divided into train/val/test — e.g. is there similar distribution of rejection grades in train vs test cohort?
- Figure S5: from which model are these predictions? Please specify it in the figure caption.